# Latent Compactness: A Unified Perspective on Generative Autoencoders from VAE to VQ-VAE

## Abstract

For a long time, the generative capability of VAE has been explained through the lens of variational inference. Conventional theories treat the KL divergence constraint and reparameterization as a unified mechanism that jointly shape the latent space. In this work, we disentangle these two components through experiments, demonstrating that the KL divergence is the key factor in fostering the formation of semantic manifolds. Reparameterization plays two roles: first, it ensures that latent representations are not deterministic points but rather anisotropic Gaussian ellipsoids, promoting a more uniform distribution in the latent space; second, it enriches the set of semantically defined points during training. This latter role is crucial for enabling meaningful sampling-based generation. Finally, we propose a unified framework where both VAE and VQ-VAE emerge as special cases. The compactness enforced by the KL divergence regularizes the latent structure, and this principle also explains why VQ-VAE, despite lacking stochasticity or a continuous prior can still achieve effective generation.

## 1 Introduction

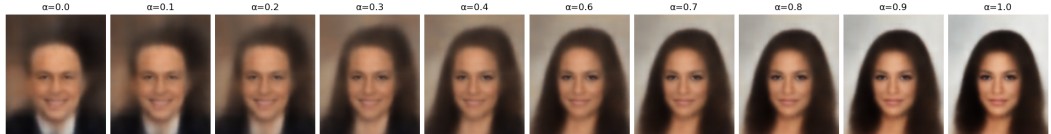

Figure 1: **10-step linear interpolations in the latent space of Autoencoder**. We use a convolutional autoencoder with latent dimension of 786, trained on 10,000 images from the CelebA dataset for 10 epochs. The figure demonstrates that the model can achieve smooth and natural transitions through linear interpolation.

Autoencoder (AE) Rumelhart et al. (1986); Baldi & Hornik (1989a) is a neural network composed of an encoder and a decoder. The encoder maps the input data $\mathbf{x}$ to a latent representation $\mathbf{z}$ through a network $\mathbf{z} = f_\theta(\mathbf{x})$. The decoder then reconstructs the input from the latent representation via $\hat{\mathbf{x}} = g_\phi(\mathbf{z})$. The goal of training an AE is to minimize the reconstruction loss, typically defined as $\mathcal{L}(\theta, \phi) = \mathbb{E}[\|\mathbf{x} - g_\phi(f_\theta(\mathbf{x}))\|^2]$.

Variational Autoencoder (VAE) Kingma & Welling (2022) extends AE by introducing a probabilistic perspective on the latent space. The training objective of VAE is to maximize the evidence lower bound (ELBO), given by $\mathcal{L}(\mathbf{x}) = \mathbb{E}_{q_\phi(\mathbf{z}|\mathbf{x})}[\log p_\theta(\mathbf{x}|\mathbf{z})] - D_{KL}(q_\phi(\mathbf{z}|\mathbf{x})\|p(\mathbf{z}))$, where $p(\mathbf{z})$ is a prior distribution (usually $\mathbf{z} \sim \mathcal{N}(\mathbf{0}, \mathbf{I})$) and $D_{KL}$ denotes the Kullback-Leibler divergence.

Although the VAE is derived from variational inference, the framework does not explain how scaling the KL divergence by a coefficient $\beta$ in $\beta$-VAE Higgins et al. (2017) allows for explicit control over the degree of disentanglement in the learned latent representation. In the ideal variational inference setting, the posterior $q_\phi(\mathbf{z}|\mathbf{x})$ is encouraged to match the prior $p(\mathbf{z}) = \mathcal{N}(\mathbf{0}, \mathbf{I})$, which would drive the KL divergence toward zero.

However, in practice, each input requires a distinct latent code, represented by its mean vector $\boldsymbol{\mu}$, while the variance $\boldsymbol{\sigma}^2$ defines a local stochastic region around it for sampling. Without this mechanism, VAEs cannot discover disentangled factors Khemakhem et al. (2020); Higgins et al. (2021) or support unsupervised learning Kingma et al. (2014).

The predecessor of VAE, AE, was long thought to lack generative capabilities. However, studies have achieved image generation using linear or circular interpolation with AE, as shown in Figure 1 ($\mathbf{z}_\alpha = (1 - \alpha)\mathbf{z}_1 + \alpha\mathbf{z}_2$) Berthelot et al. (2018). This phenomenon was subsequently explained through the lens of manifold learning Lee (2023). During interpolation, compared to random sampling, moving within the defined region of the manifold allows for smooth transitions. Nevertheless, in rare cases, the interpolation may unexpectedly move outside this region to undefined points, leading to breakdowns in AE interpolation Bengio et al. (2013b).

The puzzling aspects of the variational inference perspective on VAE, combined with the generative capabilities of autoencoders, lead us to question: is it possible to explain VAE without relying on the framework of variational inference?

In this study, we aim to disentangle the contributions of the two core components of VAE to generative capability: KL divergence and reparameterization. We posit that if structured perturbations applied to latent codes result in smooth and continuous changes in the decoder's output images, then the semantic space has been properly established and can be effectively exploited for image generation through appropriate methods. We argue that the KL divergence plays a key role in shaping a well-organized semantic space, while reparameterization serves two complementary functions: it regularizes the latent space and enriches the set of semantically defined points, thereby supporting stable and meaningful generation.

All conclusions are based on direct observation of reconstruction behavior and monitoring of purpose-designed quantitative metrics developed specifically for this investigation. We do not report conventional evaluation metrics such as sharpness or diversity, as they offer limited insight into the specific mechanisms we aim to understand. Furthermore, given that the underlying mechanisms are not yet well understood, our conclusions are based entirely on empirical observations from experiments across multiple datasets, rather than on theoretical derivation.

The Adam optimizer is used with a learning rate of 1e-3 across all experiments.

## 2 REPARAMETERIZATION AS A REGULARIZATION MECHANISM

### 2.1 GAUSSIAN BALLS MAKE THE LATENT SPACE MORE PERMISSIVE UNDER KL DIVERGENCE CONSTRAINT

In the reparameterization operation of VAE, we express the latent variable as:

$$\boldsymbol{z} = \boldsymbol{\mu} + \boldsymbol{\sigma} \odot \boldsymbol{\epsilon} \quad \boldsymbol{\epsilon} \sim \mathcal{N}(0, I)$$

where $\mu$ and $\sigma$ are the mean and standard deviation output by the encoder respectively.

We introduce the concept of a *Gaussian ball*, as shown in Figure 2a (for visualization purposes only). Under reparameterization, each latent code is no longer represented as a deterministic point, but defines a stochastic neighborhood (an anisotropic ellipsoidal region in the latent space) governed by the encoder's outputs. This ellipsoid is centered at $\boldsymbol{\mu}$ with its extent along each axis determined by the corresponding entry in the variance vector $\boldsymbol{\sigma}^2$.

As the number of reparameterized samples increases, the decoder learns not to rely on a single deterministic code, but instead to reconstruct faithfully from a neighborhood around $\hat{\boldsymbol{\mu}}$, effectively treating the encoding as an open set centered at $\boldsymbol{\mu}$. During training, the KL divergence term encourages all $\boldsymbol{\mu}$ to concentrate toward the origin, while simultaneously preventing latent codes from collapsing into a single cluster by maintaining a minimum spread around each mean. Crucially, reparameterization enforces that reconstructions remain accurate not only at $\boldsymbol{\mu}$, but also within its local vicinity, thereby shaping a latent topology filled with overlapping or adjacent Gaussian balls. Although latent codes may exhibit non-uniform distributions in practice, the VAE framework implicitly regularizes the size of these Gaussian balls through the KL divergence, which pulls the $\boldsymbol{\sigma}^2$ toward those of the unit Gaussian prior. This regularization promotes a more uniform and well-disentangled latent geometry, where codes are neither too dispersed nor overly clustered.

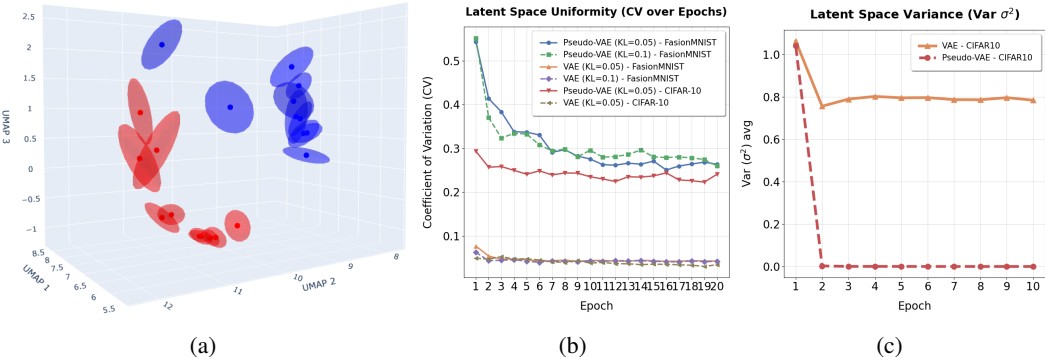

Figure 2: a): Visualization of Gaussian balls in a VAE trained on MNIST. For the test samples, each encoded distribution parameterized by mean $\boldsymbol{\mu}$ and diagonal variance $\boldsymbol{\sigma}^2 = \exp(\log \boldsymbol{\sigma}^2)$ (standard in VAE) is interpreted as a high-dimensional ellipsoid. Blue and red ellipsoids represent 10 randomly selected samples from two different classes. We use UMAP to visualize their geometry in 3D. b): Coefficient of Variation (CV) of nearest-neighbor distances in the latent space, evaluated on CIFAR-10 and FashionMNIST under different $\beta$-KL constraints ($\beta = 0.05, 0.1$) over 20 epochs. Results are averaged over 16 batches of 128 test samples each, with 1,000 random samples drawn per batch to estimate stochasticity introduced by reparameterization. c): Evolution of the average output variance $\mathbb{E}[\boldsymbol{\sigma}^2]$ across training epochs for a VAE without variance constraint and a VAE on CIFAR-10 ($\beta = 1$). The experiments use fully connected architectures for MNIST variants and custom convolutional networks for CIFAR-10.

To verify our hypothesis, we introduce a metric based on the Coefficient of Variation (CV) of nearest-neighbor distances in the latent space. For each encoded point $\mathbf{z}_i$ in a test batch, we compute its squared Euclidean distance to the nearest other point in the same batch: $d_i = \min_{j \neq i} \|\mathbf{z}_i - \mathbf{z}_j\|^2$. This yields a set of nearest-neighbor distances $\{d_i\}_{i=1}^N$. We then compute the CV of this distribution as $\mathrm{CV} = s/\bar{d}$, where $s$ is the sample standard deviation and $\bar{d}$ is the mean of the $d_i$. The CV is dimensionless and normalizes variability relative to the mean, enabling fair comparison across models with different latent space scales. A lower CV indicates more uniform distribution of points, while a higher CV suggests clustering or irregular spacing.

We conducted an experiment comparing a standard VAE that removes the variance regularization term from the KL divergence and monitored the evolution of their CV values, as shown in Figure 2b. Across different datasets and different beta coefficients for the KL divergence, the VAE without variance constraints consistently exhibits a higher CV than the standard VAE. We choose CV over raw variance because it provides better scale invariance, clearly indicating that constraining the variance (controlling the radius of the Gaussian balls) leads to a more uniform latent space. We also tracked the average output variance over training epochs, as shown in Figure 2c, where for the unconstrained VAE group, the variance rapidly collapses to near-zero by the second epoch, causing the Gaussian balls to degenerate into deterministic points.

## 2.2 Reformulated VAE Training by using Gaussian Ball assumption

We previously argued that under the KL divergence constraint, reparameterization effectively constructs a Gaussian ball, endowing each encoded point with volumetric extent. Building on this geometric interpretation, we propose two modified models that incorporate variance regularization, not as a consequence of variational inference but as an explicit mechanism to control the scale and shape of these Gaussian balls in the latent space. Both models preserve the ability to generate samples through latent space sampling.

**Asymmetric Expansion**

$$\mathcal{L}_{\text{spread}} = \sum_i \mathrm{ReLU}(-\log \boldsymbol{\sigma}_i^2)$$

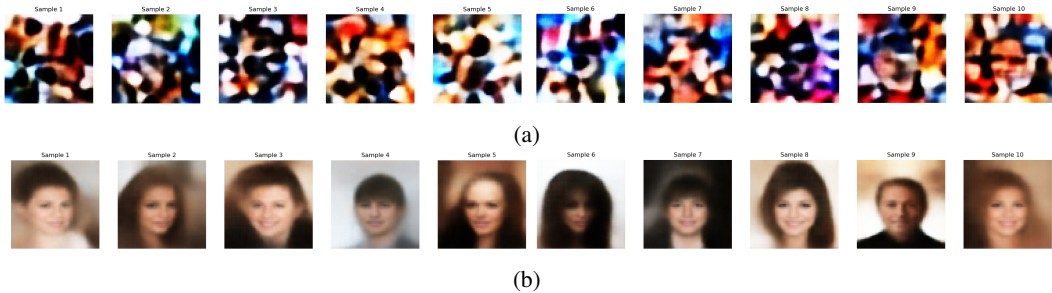

(a)

(b)

Figure 3: Comparison of image generation via random latent sampling from the prior $\mathbf{z} \sim \mathcal{N}(\mathbf{0}, \mathbf{I})$ on a subset of CelebA. Models are trained for 10 epochs. a: A baseline model with the same configuration as the VAE with variance constraint removed, as mentioned previously. b: Our proposed asymmetric regularization method.

Here, $\log \boldsymbol{\sigma}_i^2$ denotes the log-variance output by the encoder for the $i$-th latent dimension, identical to the standard VAE setup. This term enforces a minimal scale: if $\log \boldsymbol{\sigma}_i^2 < 0$ ($\boldsymbol{\sigma}_i^2 < 1$), a penalty is applied; otherwise, no constraint is imposed. This design reflects the principle: do not collapse, but grow freely, prioritizing diversity over uniformity.

**Symmetric Stabilization**

$$\mathcal{L}_{\text{spread}} = \sum_i \log \left( 1 + (\log \boldsymbol{\sigma}_i^2)^2 \right)$$

This form penalizes deviations of $\log \boldsymbol{\sigma}_i^2$ from zero in both directions, encouraging the encoded variances to remain close to unity. It's the same constraint as KL divergence, but with a different implementation.

In both cases, the full loss includes a centering term and reconstruction:

$$\mathcal{L} = \underbrace{\mathcal{L}_{\text{recon}}(x, \hat{x})}_{\text{reconstruction}} + \underbrace{\|\boldsymbol{\mu}\|^2}_{\text{centering}} + \underbrace{\mathcal{L}_{\text{spread}}}_{\text{spread regularization}}$$

For comparison, we also write the standard VAE's KL divergence with a $\mathcal{N}(0, I)$ prior:

$$\mathcal{L}_{\text{KL}} = \frac{1}{2} \sum_i \left( \mu_i^2 + \sigma_i^2 - \log \sigma_i^2 - 1 \right), \quad \sigma_i^2 = \exp(\log \text{Var}_i)$$

This term simultaneously pulls $\mu_i \to 0$ and $\sigma_i^2 \to 1$.

Both versions can achieve the same results as VAE. Here, we present results only for the **Asymmetric Expansion** version. As shown in Figure 3b, models trained with this asymmetric regularization successfully generate meaningful samples via randomly sampling. In contrast, when the spread regularization is removed and only the centering term is kept, the reconstructed images from sampling become fragmented and meaningless (see Figure 3a).

## 2.3 REPARAMETERIZATION AS A CONTRACTUAL MECHANISM FOR EXPANDING THE SET OF DEFINED POINTS

In the previous section, we discussed how reparameterization in VAE introduces Gaussian balls that regularize the distribution of encoded points, promoting greater uniformity and preventing collapse.

However, reparameterization does more than just regularize the geometry of the latent space. It effectively increases the density of actively used regions in the latent space. This implies that during training, the model is exposed to a broader set of latent configurations, even from the same input.

This raises a critical question: Is the generative capability of VAE solely due to the regularization effect of these Gaussian balls, or does it also stem from the increased density of semantic definition that more regions of the space are actively used?



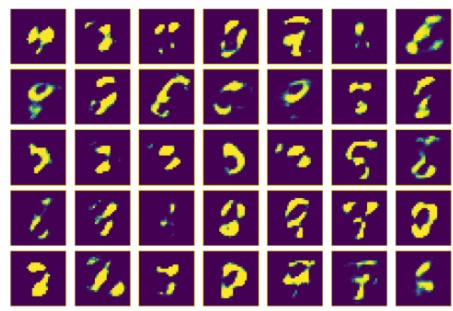

(a) Additive perturbations on the first five dimensions of one encoded sample

(b) Reparameterized samples from five training inputs

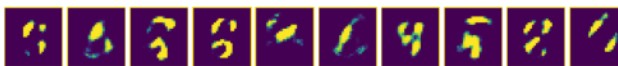

(c) A random sampling experiment on MNIST using an MLP, drawing from a Gaussian distribution.

Figure 4: Decoder replacement experiment on MNIST using an MLP architecture.

We design the following experiment to investigate: We first train a standard VAE with an MLP architecture on MNIST. After training, the decoder is removed and replaced with a randomly initialized decoder. The encoder parameters are then frozen and the mean output $\boldsymbol{\mu}$ is used as the deterministic latent code to train the new decoder, without applying reparameterization during this phase.

After training, we observe that reparameterization near the encoded vector by sampling from $\mathcal{N}(\boldsymbol{\mu}, \boldsymbol{\sigma}^2)$ fails to generate meaningful images (Figure 4b). Similarly, sampling latent codes directly from a standard Gaussian prior, $\mathbf{z} \sim \mathcal{N}(0, I)$, produces incoherent and meaningless outputs, indicating that the newly trained decoder does not maintain compatibility with the standard normal prior (Figure 4c). Surprisingly, however, when we apply fixed additive perturbations along individual latent dimensions ( $z_i \leftarrow \mu_i + \delta$ ) for a fixed $\delta$ the decoder produces smooth and semantically meaningful transitions, despite the absence of stochastic sampling or KL regularization during training (Figure 4a).

This demonstrates that reparameterization does more than just regularize the latent space: it effectively enriches the set of defined points, acting as a contractual mechanism between the encoder and decoder. Without this enforced expansion of coverage around each code, the decoder alone cannot interpret the latent structure; as a result, it fails to generate meaningful samples through random sampling. Yet, the observation that smooth transitional reconstructions are still possible under fixed perturbations (unattainable in standard autoencoder) highlights the critical contribution of the KL divergence in shaping a compact latent space that supports semantic continuity. We will elaborate on this mechanism in the next section.

## 3 EXTENDING THE GEOMETRIC VIEW: CONNECTION TO VQ-VAE

### 3.1 THE ROLE OF KL DIVERGENCE IN LATENT SPACE FILLING

In this section, we will examine the influence of KL divergence itself and its effect on filling the latent space.

We introduce *Dynamic Latent Coverage*, a metric that tracks how well the encoder's representations cover the standard normal prior over the course of training. Inspired by prior work on representation sparsity Cheung et al. (2014), geometric structure in latent spaces Bengio et al. (2013c), and detection of underutilized regions in VAE Rubenstein et al. (2018), our method provides a temporal view of latent space utilization by measuring the proportion of prior samples that fall within high-density neighborhoods of real data encodings.

The metric is computed at the end of each training epoch using a fixed evaluation set and an adaptive neighborhood radius, as detailed in Algorithm 1.

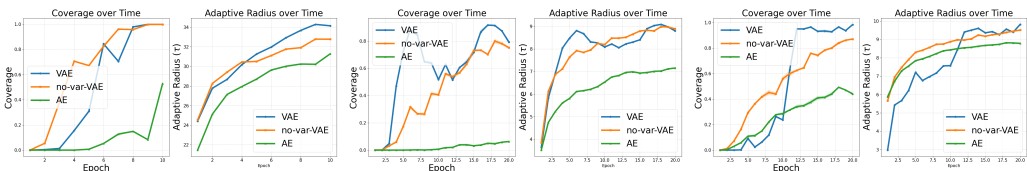

Figure 5: Dynamic Latent Coverage and adaptive radius $\tau$ during training on CelebA (left), FashionMNIST (middle), and MNIST (right).The CelebA model employs a custom convolutional architecture, while FashionMNIST and MNIST use feedforward networks. Training runs for 10 epochs on CelebA and 20 epochs on the MNIST variants. The hyperparameters follow those defined in Algorithm 1. Both metrics are computed at the end of each epoch using a fixed evaluation set.

Here, we measure the evolution of our proposed coverage metric over epochs for the standard VAE, the VAE without variance constraints (no-var-VAE), and a standard autoencoder (AE), across different datasets, as shown in Figure 5. We first observe that $\tau$ changes across epochs, which justifies the necessity of our adaptive radius design. Over the course of training, the coverage increases for all models, but the models with KL regularization (VAE&no-var-VAE) exhibit faster growth compared to the standard AE.

This phenomenon is intriguing: for the standard VAE, the space coverage metric grows most rapidly, and even shows a sudden jump in MNIST and FashionMNIST. In contrast, the no-var-VAE exhibits slower but gradual improvement. This observation, combined with the finding in Figure 4a where replacing the decoder with a newly trained one still yields meaningful responses to perturbations raises an important question: Could it be that reparameterization primarily serves as a mechanism to enrich the set of defined points, enabling sample-based generation, while the centralizing term $\|\boldsymbol{\mu}\|^2$ in the KL divergence is actually the key factor driving the formation of a structured semantic space and enabling meaningful generative capabilities?

This is what we aim to investigate in the following section.

### 3.2 MODEL ARCHITECTURE

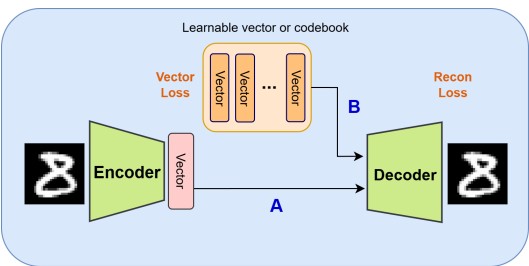

Figure 6: Model Architecture. Path A: for our experiment, where the encoder output is regularized toward the nearest code in a learnable codebook. Path B: VQ-VAE, which uses discrete latent codes.

We now introduce the modelsto be used in the following experiments, as illustrated in Figure 6. We train an autoencoder with a learnable codebook $\mathcal{C} = \{c_1, \ldots, c_K\}$. For a given input $x$, the encoder produces $\mu = f_{\text{enc}}(x)$, and the nearest code is found as $k^* = \arg\min_k \|\mu - c_k\|_2$. The loss combines reconstruction and codebook regularization:

$$\mathcal{L} = \|x - f_{\text{dec}}(\mu)\|_2^2 + \lambda \|\mu - c_{k^*}\|_2^2.$$

This encourages the latent space to form compact clusters around learnable centers. While constraining $\mu$ toward zero would also promote compactness, a learnable codebook allows richer geometric structures. In the figure, path A corresponds to our method, and path B to VQ-VAE, establishing a conceptual connection: both approaches regularize the latent space to shape meaningful semantic

manifolds, albeit through different mechanisms. Notably, VQ-VAE can be viewed as a special case of our framework, where the latent representation is replaced with the code $c_{k*}$ during decoding.

### 3.3 TRAINING AUTOENCODERS VIA COMPACTNESS REGULARIZATION

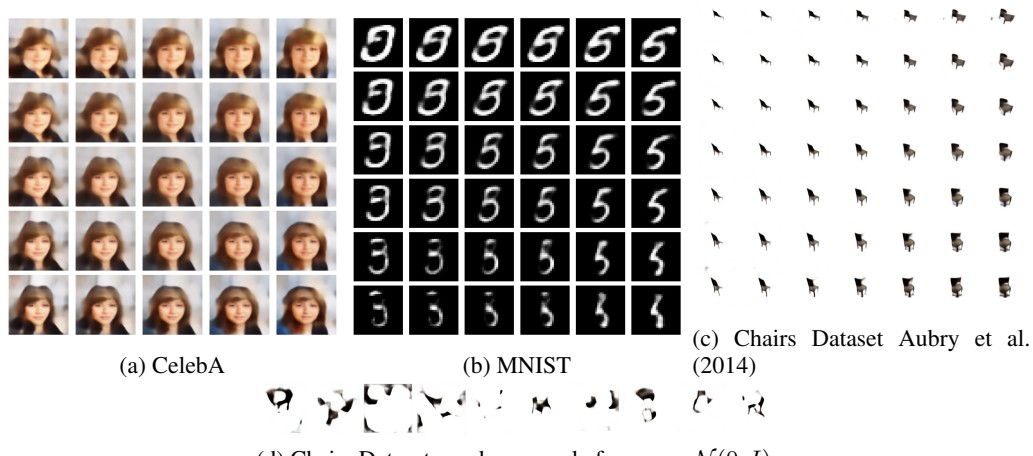

(a) CelebA           (b) MNIST           (c) Chairs Dataset Aubry et al. (2014)

(d) Chairs Dataset, random sample from $z \sim \mathcal{N}(0, I)$.

Figure 7: Top row: Dimensional perturbation analysis on autoencoders regularized with a learnable vector ( $\lambda = 0.1$ ). Each panel shows decoder outputs when perturbing each latent dimension by $[-3, -2, -1, 1, 2, 3]$ (horizontal axis), with dimensions along the vertical axis. Models are trained for 15 epochs. Bottom: Generated samples from $z \sim \mathcal{N}(0, I)$, passed through the decoder.

Here we discuss the case of constraining to a single vector, temporarily setting aside the idea of a codebook. For the sampling experiments, this vector is fixed at zero rather than being learnable, effectively resulting in an autoencoder regularized by the $\|\boldsymbol{\mu}\|^2$ term.

As shown in Figure 7, the autoencoder with codebook regularization exhibits smooth and meaningful responses to dimensional perturbations across all three datasets, consistent with the observations in Figure 4. This suggests that the compactness enforced by the KL divergence in the latent space that enables continuous transitions along dimensions. Meanwhile, reconstructions from sampling $z \sim \mathcal{N}(0, 1)$ (Figure 7d) are fragmented and meaningless, consistent with the observations in Figure 4 where the decoder was replaced. This indicates that reparameterization enables image generation from random samples, while the smooth transitions along individual dimensions arise from the compact latent space structure imposed by the KL divergence.

### 3.4 EXTENDING THE CONSTRAINT FROM A SINGLE VECTOR TO THE CODEBOOK

To further investigate the underlying dynamics and move closer to VQVAE, we extend the constraint from a single vector to the entire codebook, as shown in Figure 8. On the MNIST dataset, each codebook entry learns a fragment of an image, serving as a kind of **cluster center**. Each vector captures a piecewise feature. Interestingly, the fourth-to-last vector becomes a blurry average image, clearly failing to learn meaningful content.

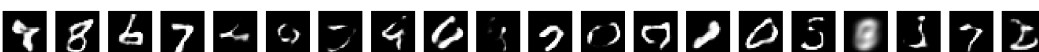

Figure 8: AE constraint experiment on MNIST trained for 20 epochs, extending the constraint from a single vector to the codebook. The figure shows the reconstruction results from 20 codebook vectors directly fed through the decoder.

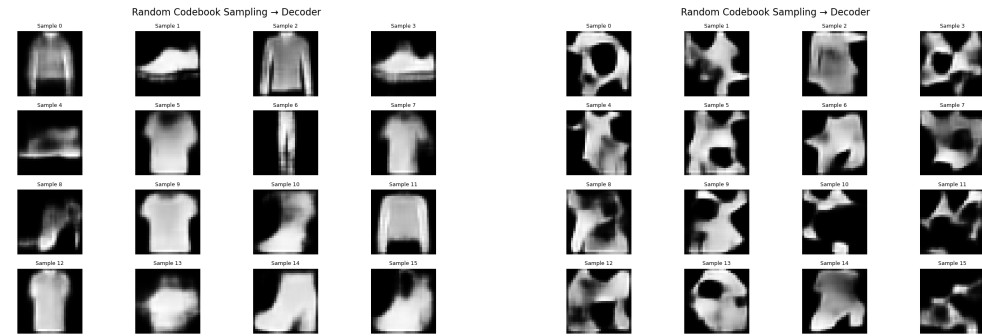

(a) Codebook with 20 vectors, encoder outputs 4 vectors.

(b) Codebook with 50 vectors, encoder outputs 20 vectors.

Figure 9: Random codebook sampling experiments on FashionMNIST using a convolutional autoencoder trained for 15 epochs. The codebook is updated via EMA to avoid collapse to identical vectors. Latent representations follow the VQ-VAE van den Oord et al. (2017) convention: feature maps of shape $[B, C, H, H]$, where each codebook vector corresponds to a channel-wise entry $[B, C, i, j]$ at spatial position $(i, j)$.

### 3.5 FROM SINGLE CODE TO CODEBOOK: SEMANTIC SPACE COLLAPSE

Next, we demonstrate how the semantic space degenerates. We then increase the number of output vectors from the encoder, as shown in Figure 9. This VQ-VAE-like approximation degenerates into a deterministic discrete encoder as we scale up the codebook size and the number of encoder output vectors. While it still correctly reconstructs inputs, whether using the encoder's representation or finding the nearest neighbor in the codebook, randomly sampling from the codebook produces meaningless, fragmented outputs. The model retains only reconstruction capability.

Although the precise underlying mechanism remains unclear and we cannot yet offer a definitive explanation, we propose a plausible hypothesis: unlike standard VQ-VAE, our reconstruction path directly passes the encoder's output to the decoder, while the codebook serves only as a constraint. When the codebook grows large, this creates an *inverse compactness constraint*, which means that effectively pulls the semantic space apart and prevents coherent learning.

Why standard VQ-VAE does not suffer from this issue may lie in the fact that its decoder receives only codebook vectors, not the raw encoder outputs. This is akin to the contractual mechanism between encoder and decoder induced by reparameterization which we previously discussed, effectively anchoring meaningful points in the latent space. However, this requires further investigation.

## 4 CONCLUSION

Our experimental results elucidate the distinct roles of the KL divergence and reparameterization. The KL divergence acts as a mechanism to enforce compactness in the latent space, enabling smooth transitions under perturbations. This partially explains why VQ-VAE, despite lacking stochasticity, can still achieve meaningful image generation. Reparameterization enriches the set of semantically defined points, effectively filling the latent space and making sampling-based generation feasible.

Moreover, the variance regularization within the KL term ensures that latent representations maintain a Gaussian ellipsoidal structure rather than collapsing to single points. This further regularizes the latent space, promoting greater uniformity. However, how exactly this geometric regularization influences generative performance remains unclear.

Despite these insights, several limitations remain. The underlying mechanisms behind our observations are not yet fully understood, and the reasons why the semantic structure breaks down as the number of encoding centers increases require further investigation. These questions call for deeper theoretical and empirical analysis in future work.

## 5 RELATED WORK

Early work on linear autoencoders established that they effectively perform PCA when trained with squared reconstruction loss Baldi & Hornik (1989b), suggesting that the learned latent space captures the primary modes of the data. This observation was later echoed in VAE, where VAE also recover the principal subspace of the data distribution Rolinek et al. (2019), further hinting at a shared geometric mechanism between classical AE and VAE despite their differing probabilistic foundations.

Numerous research explored how standard AE can be endowed with generative capabilities, including regularizing the latent space with topological constraints Lee (2023); Connor et al. (2021); Leeb et al. (2023), enforcing smoothness via contractive or denoising objectives Rifai et al. (2011b), or sampling from neighborhoods around encoded points Bengio et al. (2013a). These efforts reflect a growing understanding that generation in latent space depends not only on stochasticity but also on the structure and regularity of the learned manifold. Indeed, several works have interpreted VAE through the lens of manifold learning Yu et al. (2025); Zheng et al. (2023); van den Berg et al. (2018).

More recently, the importance of compactness has been highlighted as a key factor for effective generation. A notable work De Bortoli et al. (2021) proposed a new generative model built upon this principle, arguing that compactness, rather than stochasticity enables reliable decoding and interpolation. Our work builds on this insight, but provides a more systematic disentanglement between the roles of compactness and reparameterization, clarifying how geometric regularization through the KL constraint shapes the latent space to support generation.

## 6 LLM DECLARATION

Large language models were used for writing refinement and literature review assistance. All other contributions including conceptualization, methodology design, experimentation, data analysis, and interpretation, were carried out by the human authors.

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

# A APPENDIX

## A.1 ALGORITHM OF DYNAMIC LATENT COVERAGE

---
**Algorithm 1** Dynamic Latent Coverage at Epoch $t$

---
**Input:** Encoder $f^\mu$, eval set $\mathcal{D} = \{x_i\}_{i=1}^M$, $M = 1000$
       Prior samples $N = 1000$, $k_{\text{nn}} = 5$, percentile $\eta$ (e.g., 95)
**Output:** Coverage$_t \in [0, 1]$

1: **for** $i = 1$ to $M$ **do**
   $\mu_i \leftarrow f^\mu(x_i)$
2: **end for**
3: $S \leftarrow \{\mu_1, \ldots, \mu_M\}$
4: $D \leftarrow \emptyset$
5: **for** $i = 1$ to $M$ **do**
   Find $k_{\text{nn}}$ nearest neighbors of $\mu_i$ in $S \setminus \{\mu_i\}$
   **for** $j = 2$ to $k_{\text{nn}}$ **do**    // skip closest
      $d \leftarrow \|\mu_i - \mu_{(j)}\|_2$; Append $d$ to $D$
   **end for**
6: **end for**
7: $\tau \leftarrow \text{percentile}(D, \eta)$
8: $c \leftarrow 0$
9: **for** $i = 1$ to $N$ **do**
   $d_{\min} \leftarrow \min_j \|\mu_j - z_i\|_2$ where $z_i \sim \mathcal{N}(0, I)$
   **if** $d_{\min} < \tau$ **then** $c \leftarrow c + 1$ **end if**
10: **end for**
11: **return** $c/N$

---

## A.2 AUTOENCODERS POSSESS GENERATIVE CAPABILITIES

As the predecessor of Variational Autoencoder (VAE), the Autoencoder (AE) has been regarded in some studies as a nonlinear extension of Principal Component Analysis (PCA). Like many deep learning models including those used in classification Rifai et al. (2011a) the AE learns a latent manifold by projecting high-dimensional data into a lower-dimensional space, thereby reshaping the structure of the original distribution Lee (2023). While AE is not conventionally treated as a generative model, prior work has suggested that it can generate new samples through latent space interpolation Berthelot et al. (2018); Sainburg et al. (2019). A similar strategy is employed in Style-GAN Karras et al. (2019; 2018). This section will further explore the generative potential of AE from this perspective.

## A.3 PERTURBATION AND INTERPOLATION OF LATENT CODES IN AUTOENCODERS

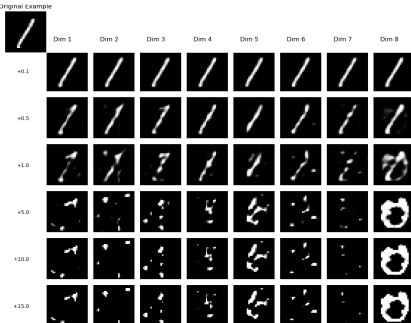

Figure 10: Add perturbation on the first 8 dimension of the latent space.From +0.1 to +15 Both the encoder and decoder are implemented as 64-dimensional MLPs, 3 layers,with a 128-dimensional latent (encoding) space.

Figure 11: Linear interpolation of number 7 and 8, 3 and 8.

As shown in Figure 10, small perturbations in the latent space of an autoencoder result in minimal changes to the decoder's output. However, under large perturbations, the generated images degrade and may even collapse into merely pure black. Next, we perform linear interpolation between the latent codes of two images Figure 11. The interpolation is defined as:

$$\mathbf{z} = (1 - \alpha)\mathbf{z}_1 + \alpha\mathbf{z}_2, \quad \alpha \in [0, 1]$$

where $\mathbf{z}_1$ and $\mathbf{z}_2$ are the latent codes of two input images, and $\alpha$ controls the interpolation weight. The decoder produces smoothly transitioning outputs, demonstrating continuous variation between the original images.

### A.4 MORE RESULTS FOR SECTION "EXTENDING THE GEOMETRIC VIEW: CONNECTION TO VQ-VAE"

#### A.4.1 TRAINING WITH MULTIPLE VECTORS

On the MNIST/FashionMNIST dataset, we observed that after 20 epochs of training, these vectors landed in different cluster centers. At this point, this constraint was relaxed, and similar to VQ-VAE, degeneration still occurred. Some of these vectors eventually converged to the same location, or simply collapsed into an identical solution, as shown in Figure 12.

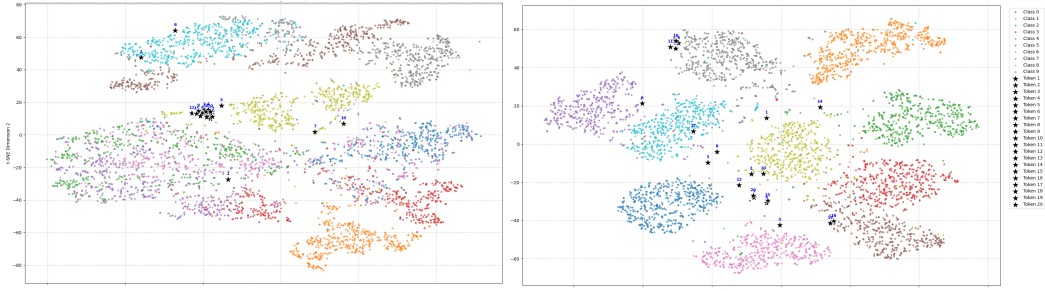

Figure 12: The left image shows a collapse scenario during training on FashionMNIST, while the right image shows normal behavior without collapse on MNIST. These two datasets were chosen to better illustrate the experimental results because FashionMNIST is more prone to collapse, whereas MNIST is difficult to collapse. **The difference in linear separability for the 10 classes in the two figures is due to the inherent nature of the datasets themselves, and not related to the model.**

We focus on using a codebook rather than a single vector for training. We previously observed that FashionMNIST is highly prone to collapse, ultimately becoming indistinguishable from a single vector.

In VQ-VAE, one of the strategies used is Exponential Moving Average (EMA)van den Oord et al. (2017). EMA reduces the oscillation of individual vectors through momentum updates, helping to prevent them from collapsing into the same solution. The core formula for updating a codebook vector $e_k$ using EMA is denoted as follows:

$$\mathbf{e}_k \leftarrow \mathbf{e}_k + \alpha(\mathbf{z}_q - \mathbf{e}_k)$$

Where: $\mathbf{e}_k$ is the $k$-th vector in the codebook. $\mathbf{z}_q$ is the latent vector. $\alpha$ is the momentum (or learning) rate.

The final t-SNE is shown in Figure 13b. The data space also becomes more flexible, as depicted in Figure 13a. Here, we applied a +300 additive perturbation, which caused a T-shirt to become longer and its sleeve length to change. Additionally, distortion also occurred.

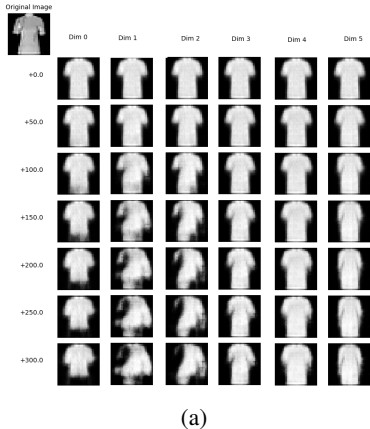

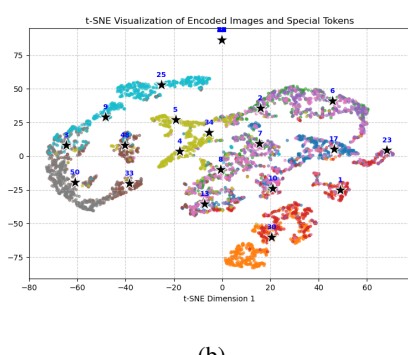

(a)                                                                 (b)

Figure 13: Experimental results on FashionMNIST. a shows the reconstructions from 50 trained vectors, while b displays the t-SNE visualization of 20 trained vectors and the encoder output on the test dataset.

### A.4.2 IMPACT OF ENCODER CAPACITY ON LEARNABLE VECTOR QUANTITY

In Figure 13b, we observed something interesting: with 50 learnable vectors, only a small number were properly utilized and attracted data.

To address this, we expanded both the encoder and decoder, making them deeper and incorporating residual connections. The results, shown in Figure 14, are promising. This time, with 500 vectors, visibly more of them were correctly "attracted" to data clusters. We then randomly selected 200 of these vectors and fed them into the decoder for visualization (Figure 15).

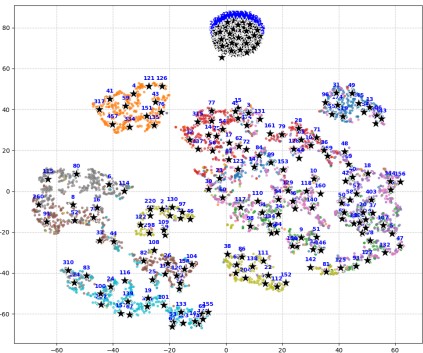

Figure 14: The t-SNE visualization displays the 500 trained vectors and the encoder output from the test dataset. Notably, a large number of these trained vectors still haven't been correctly attracted to data clusters.

### A.4.3 MORE RESULTS FOR SECION "FROM SINGLE CODE TO CODEBOOK: SEMANTIC SPACE COLLAPSE"

We present additional experiments on collapse.

From Figure 18, we observe that at this point, no matter how much perturbation is added, the model does not gain a transition capability. Instead, it only increases distortion. In Figure 17, completely random combinations yield only meaningless results. In Figure 20, arbitrarily replacing one of the

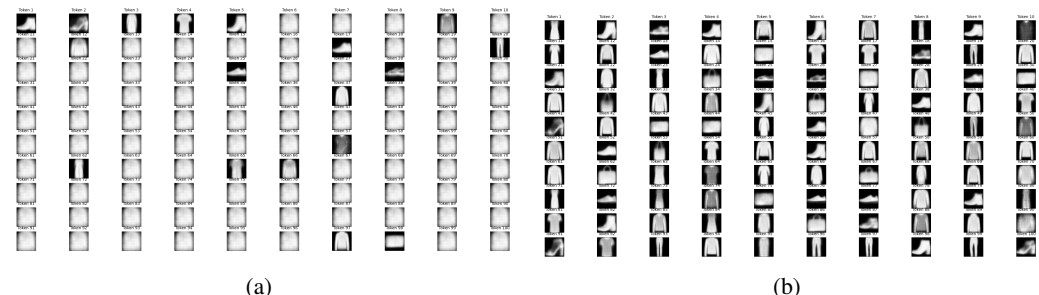

(a)                                                                (b)

Figure 15: Experimental results on FashionMNIST with an expanded model. a shows reconstructions from randomly selected 100 correctly learned vectors. b displays reconstructions from many incorrectly learned vectors, where numerous meaningless white ones are present.

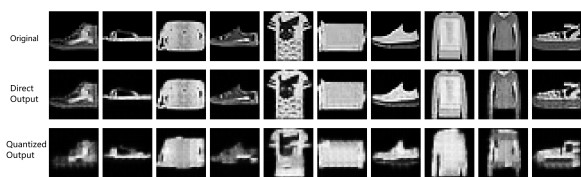

Figure 16: From top to bottom: Original Image, Encoder-Decoder Direct Output, Quantized Output (Codebook to Decoder)

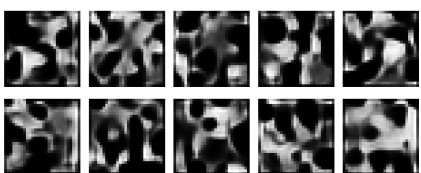

Figure 17: Reconstruction of images generated by randomly combining vectors sampled from the codebook and fed into the decoder.

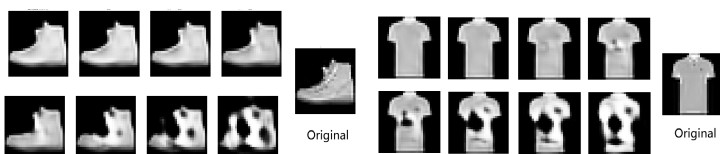

Figure 18: Additive perturbations of +1, +5, +10, +15, +30, +50, and +100 to the first dimension of the first vector in the encoder-to-decoder output.

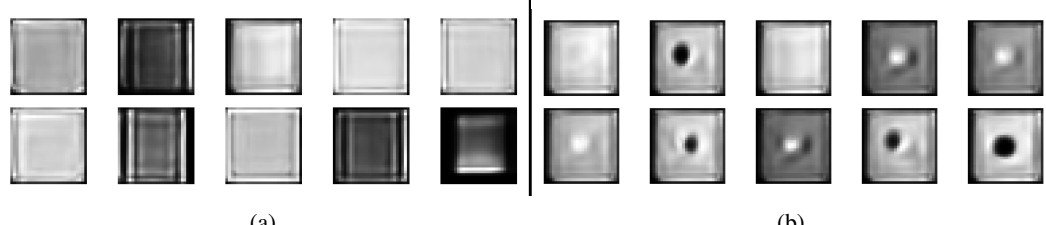

(a)                                                                (b)

Figure 19: Reconstruction results using codebook vectors passed to the decoder via broadcasting a and zero-padding b.

Figure 20: Reconstruction after randomly replacing the encoder's output with a random codebook vector.

vectors with another from the codebook also introduces distortion. In Figure 19, the reconstruction results of the codebook vectors themselves are indistinguishable. In Figure 16, it is visible that although the model's codebook vectors cannot learn meaningful semantics like in VQ-VAE, they form the original image through correct combinations, and these are indeed random combinations, not a degeneration into an AE. (The codebook indices used for this set of images are in Appendix A.4.3)

Therefore, we can conclude that in such circumstances, the model tends to become a discrete encoder. Although each vector in the codebook lacks semantic meaning, it can generate relatively clear original images through combination.

## A.5 CODEBOOK INDICES USED IN THE EXPERIMENT

In our experiment, the encoder's output convolution map has dimensions of 7x7 with 512 channels. Below are the codebook indices matched to each convolutional map location (with channels unfolded as vectors, consistent with VQ-VAE operations).

$$
\begin{bmatrix}
9 & 10 & 10 & 10 & 10 & 10 & 10 \\
47 & 44 & 44 & 44 & 72 & 64 & 37 \\
47 & 44 & 44 & 26 & 30 & 96 & 32 \\
47 & 26 & 72 & 25 & 89 & 11 & 94 \\
18 & 71 & 69 & 3 & 3 & 1 & 8 \\
19 & 14 & 20 & 16 & 16 & 16 & 87 \\
19 & 7 & 7 & 37 & 50 & 37 & 37
\end{bmatrix}
\quad
\begin{bmatrix}
9 & 17 & 21 & 31 & 36 & 79 & 10 \\
47 & 58 & 24 & 60 & 3 & 95 & 58 \\
47 & 44 & 54 & 89 & 23 & 37 & 58 \\
47 & 44 & 59 & 60 & 57 & 38 & 58 \\
47 & 44 & 59 & 60 & 74 & 38 & 58 \\
47 & 44 & 59 & 60 & 57 & 38 & 58 \\
19 & 7 & 54 & 94 & 57 & 38 & 7
\end{bmatrix}
$$

$$
\begin{bmatrix}
9 & 17 & 21 & 80 & 31 & 79 & 17 \\
19 & 25 & 0 & 65 & 65 & 0 & 6 \\
91 & 40 & 65 & 0 & 0 & 89 & 53 \\
91 & 40 & 65 & 0 & 0 & 89 & 93 \\
28 & 69 & 69 & 0 & 0 & 33 & 81 \\
28 & 1 & 69 & 65 & 0 & 42 & 2 \\
91 & 32 & 33 & 1 & 0 & 33 & 81
\end{bmatrix}
\quad
\begin{bmatrix}
9 & 17 & 85 & 36 & 36 & 79 & 10 \\
47 & 26 & 27 & 0 & 22 & 64 & 58 \\
47 & 26 & 27 & 0 & 0 & 64 & 58 \\
47 & 26 & 27 & 41 & 11 & 64 & 58 \\
47 & 26 & 30 & 66 & 11 & 6 & 58 \\
47 & 26 & 30 & 81 & 0 & 6 & 58 \\
19 & 26 & 30 & 93 & 80 & 6 & 7
\end{bmatrix}
$$

$$
\begin{bmatrix}
9 & 17 & 17 & 10 & 17 & 79 & 10 \\
19 & 29 & 23 & 82 & 59 & 32 & 6 \\
91 & 43 & 3 & 3 & 65 & 3 & 66 \\
73 & 60 & 68 & 65 & 65 & 65 & 74 \\
73 & 60 & 68 & 68 & 65 & 65 & 94 \\
73 & 60 & 68 & 60 & 68 & 65 & 94 \\
76 & 39 & 39 & 39 & 39 & 39 & 75
\end{bmatrix}
\quad
\begin{bmatrix}
9 & 45 & 77 & 36 & 36 & 77 & 17 \\
19 & 35 & 35 & 35 & 95 & 35 & 50 \\
47 & 70 & 62 & 35 & 35 & 35 & 50 \\
47 & 26 & 35 & 87 & 87 & 64 & 58 \\
47 & 26 & 35 & 87 & 87 & 64 & 58 \\
47 & 7 & 35 & 87 & 87 & 35 & 58 \\
19 & 72 & 35 & 87 & 87 & 87 & 37
\end{bmatrix}
$$

$$
\begin{bmatrix}
9 & 10 & 45 & 36 & 5 & 10 & 10 \\
47 & 44 & 72 & 80 & 66 & 58 & 58 \\
47 & 44 & 72 & 0 & 66 & 58 & 58 \\
47 & 44 & 70 & 0 & 49 & 37 & 58 \\
47 & 44 & 29 & 65 & 32 & 38 & 58 \\
47 & 26 & 30 & 65 & 3 & 6 & 58 \\
19 & 7 & 40 & 1 & 1 & 34 & 7
\end{bmatrix}
\quad
\begin{bmatrix}
9 & 10 & 17 & 77 & 77 & 17 & 10 \\
47 & 44 & 72 & 66 & 4 & 6 & 58 \\
47 & 26 & 29 & 11 & 84 & 49 & 37 \\
19 & 59 & 3 & 65 & 65 & 3 & 66 \\
91 & 67 & 3 & 65 & 89 & 3 & 81 \\
28 & 55 & 3 & 65 & 65 & 65 & 2 \\
19 & 25 & 94 & 94 & 96 & 96 & 93
\end{bmatrix}
$$

$$
\begin{bmatrix}
9 & 10 & 10 & 10 & 10 & 10 & 10 \\
47 & 44 & 44 & 26 & 7 & 44 & 26 \\
47 & 26 & 7 & 71 & 67 & 86 & 71 \\
18 & 71 & 84 & 0 & 0 & 0 & 63 \\
51 & 80 & 0 & 56 & 63 & 63 & 22 \\
19 & 50 & 12 & 12 & 14 & 14 & 14 \\
19 & 7 & 7 & 7 & 7 & 7 & 7
\end{bmatrix}
\quad
\begin{bmatrix}
9 & 10 & 21 & 80 & 49 & 17 & 10 \\
47 & 26 & 4 & 22 & 80 & 6 & 58 \\
47 & 44 & 4 & 32 & 80 & 38 & 58 \\
47 & 44 & 70 & 32 & 80 & 38 & 58 \\
47 & 44 & 72 & 32 & 22 & 38 & 58 \\
47 & 44 & 72 & 32 & 27 & 38 & 58 \\
19 & 7 & 72 & 49 & 32 & 38 & 7
\end{bmatrix}
$$

$$
\begin{bmatrix}
9 & 10 & 10 & 10 & 10 & 17 & 45 \\
47 & 44 & 26 & 86 & 71 & 84 & 75 \\
47 & 26 & 4 & 1 & 60 & 55 & 78 \\
19 & 4 & 63 & 75 & 3 & 22 & 80 \\
67 & 56 & 8 & 56 & 22 & 32 & 80 \\
73 & 80 & 80 & 80 & 62 & 27 & 22 \\
91 & 12 & 14 & 14 & 50 & 14 & 46
\end{bmatrix}
\quad
\begin{bmatrix}
9 & 45 & 36 & 22 & 27 & 31 & 17 \\
19 & 29 & 11 & 0 & 0 & 1 & 34 \\
19 & 27 & 11 & 63 & 22 & 0 & 81 \\
91 & 69 & 11 & 22 & 22 & 0 & 2 \\
28 & 0 & 65 & 69 & 80 & 0 & 49 \\
28 & 0 & 65 & 0 & 0 & 0 & 32 \\
76 & 74 & 39 & 39 & 24 & 63 & 32
\end{bmatrix}
$$

$$
\begin{bmatrix}
9 & 17 & 85 & 13 & 83 & 79 & 10 \\
47 & 59 & 3 & 68 & 68 & 23 & 38 \\
47 & 29 & 74 & 65 & 65 & 74 & 6 \\
19 & 30 & 68 & 65 & 65 & 60 & 34 \\
19 & 43 & 60 & 68 & 68 & 89 & 53 \\
91 & 40 & 69 & 60 & 60 & 42 & 93 \\
91 & 43 & 46 & 46 & 46 & 30 & 53
\end{bmatrix}
\quad
\begin{bmatrix}
9 & 17 & 17 & 17 & 17 & 17 & 17 \\
19 & 4 & 87 & 67 & 67 & 78 & 78 \\
91 & 27 & 22 & 22 & 0 & 0 & 0 \\
76 & 22 & 22 & 0 & 0 & 0 & 0 \\
73 & 27 & 80 & 0 & 0 & 0 & 80 \\
73 & 22 & 22 & 63 & 22 & 22 & 24 \\
91 & 37 & 50 & 50 & 37 & 37 & 37
\end{bmatrix}
$$

$$
\begin{bmatrix}
9 & 10 & 10 & 10 & 10 & 10 & 10 \\
47 & 44 & 44 & 7 & 7 & 26 & 7 \\
47 & 26 & 7 & 25 & 80 & 78 & 66 \\
19 & 70 & 80 & 30 & 0 & 0 & 16 \\
18 & 43 & 63 & 32 & 63 & 1 & 80 \\
51 & 46 & 20 & 20 & 46 & 16 & 46 \\
19 & 7 & 7 & 7 & 7 & 7 & 7
\end{bmatrix}
\quad
\begin{bmatrix}
9 & 45 & 85 & 13 & 83 & 5 & 17 \\
19 & 43 & 3 & 65 & 8 & 3 & 34 \\
91 & 56 & 3 & 68 & 65 & 8 & 53 \\
47 & 50 & 69 & 68 & 68 & 93 & 50 \\
47 & 58 & 69 & 68 & 68 & 93 & 58 \\
47 & 72 & 69 & 68 & 68 & 93 & 58 \\
19 & 72 & 1 & 94 & 94 & 93 & 7
\end{bmatrix}
$$

$$
\begin{bmatrix}
9 & 10 & 10 & 10 & 10 & 10 & 10 \\
47 & 44 & 44 & 26 & 86 & 37 & 86 \\
47 & 44 & 26 & 72 & 48 & 69 & 32 \\
19 & 26 & 86 & 87 & 22 & 89 & 0 \\
18 & 87 & 48 & 80 & 80 & 1 & 80 \\
51 & 14 & 20 & 20 & 20 & 16 & 20 \\
19 & 7 & 7 & 7 & 7 & 7 & 7
\end{bmatrix}
\quad
\begin{bmatrix}
9 & 17 & 17 & 17 & 17 & 17 & 17 \\
18 & 71 & 71 & 71 & 71 & 71 & 78 \\
80 & 0 & 0 & 0 & 0 & 0 & 1 \\
40 & 69 & 8 & 69 & 69 & 8 & 96 \\
69 & 0 & 0 & 0 & 0 & 0 & 1 \\
69 & 1 & 0 & 1 & 1 & 1 & 1 \\
51 & 14 & 14 & 14 & 12 & 12 & 12
\end{bmatrix}
$$

$$
\begin{bmatrix}
9 & 10 & 17 & 56 & 2 & 17 & 10 \\
47 & 44 & 72 & 89 & 57 & 37 & 58 \\
47 & 44 & 72 & 89 & 57 & 38 & 58 \\
47 & 44 & 70 & 65 & 1 & 6 & 58 \\
47 & 44 & 59 & 65 & 0 & 6 & 58 \\
47 & 44 & 54 & 65 & 74 & 38 & 58 \\
19 & 7 & 54 & 1 & 57 & 38 & 7
\end{bmatrix}
\quad
\begin{bmatrix}
9 & 10 & 45 & 13 & 36 & 5 & 10 \\
47 & 44 & 72 & 89 & 60 & 53 & 58 \\
47 & 44 & 54 & 89 & 68 & 34 & 58 \\
47 & 44 & 54 & 60 & 60 & 34 & 58 \\
47 & 44 & 59 & 60 & 68 & 53 & 58 \\
47 & 44 & 29 & 60 & 60 & 53 & 58 \\
19 & 7 & 25 & 16 & 16 & 12 & 58
\end{bmatrix}
$$