# OpenReview forum: "Latent Compactness: A Unified Perspective on Generative Autoencoders from VAE to VQ-VAE"
_ICLR.cc/2026/Conference — ICLR 2026 Conference Withdrawn Submission_

### Official Review · Reviewer_C9Di · 2025-10-24

**Soundness:** 3
**Presentation:** 3
**Contribution:** 2
**Rating:** 2
**Confidence:** 4

**Summary:**

This paper presents a conceptual reinterpretation of Variational Autoencoders (VAEs), aiming to disentangle the distinct functional roles of the Kullback–Leibler (KL) divergence and the reparameterization trick within the standard VAE formulation. Contrary to the conventional view that treats these mechanisms as merely components of variational inference, the authors argue that they are structurally complementary: the KL divergence enforces latent compactness and semantic organization, while reparameterization ensures latent enrichment and sample diversity.
Empirical analyses support these interpretations. The authors propose alternative VAE variants with explicit Gaussian-ball regularization extracted from the encoder (mean and variance of the variational inference distribution). They also experiment the link between the joint training of encoder, latent representation and decoder showing their interdependence. They compare with VQ-VAE which falls in the same category of models.

**Strengths:**

The importance of the joint model training is very well explained and experiments support this fact.
The geometrical representation of the latent space and the link (given the ball-parametrisation) with VQ-VAE is interesting.

**Weaknesses:**

One paper on the exact same viewpoint lacks in the related works and comparison with this particular paper would have help.
A Geometric Perspective on Variational Autoencoders. Chadebec, C. and Allassonnière, S. Neural Information Processing Systems (NeurIPS 2022).

Results are only qualitative. Quantitative analysis is lacking.

**Questions:**

Since you consider the geometric structure of the latent space, why do you perform linear interpolation?
What is the impact of the latent dimension? In particular when considering the latent balls, does this representation enables to compact the information into lower dimensional space?
When computing the nearest neighbours, you use the Euclidean distant whereas you have local anisotropy. Is the anisotropic distance performing better/worse?

---

### Official Review · Reviewer_R8r4 · 2025-10-28

**Soundness:** 2
**Presentation:** 2
**Contribution:** 2
**Rating:** 4
**Confidence:** 3

**Summary:**

This paper studies distinct roles of the KL divergence and reparameterization in VAEs and argues that the KL term is primarily responsible for forming a well-organized latent space. It proposes a unified framework where both VAE and VQ-VAE can be seen as special cases of latent compactness regularization. Their findings are supported through empirical experiments on multiple datasets in vision.

**Strengths:**

1. The paper is clearly structured and is easy to follow. The intuitive illustrations are helpful.
2. The authors present ablation studies systematically. These ablations help isolate the contributions of each component.

**Weaknesses:**

1. The main insights have been discussed extensively in prior works and are well-known to VAE users. For instance, the following VAE tutorial shows the same idea: https://avandekleut.github.io/vae/
2. If the paper wants to study this phenomenon deeply, then it should contain rigorous statements and derivations. Currently the claims are qualitative. For instance, when introducing Gaussian balls, why could each code represent a stochastic neighborhood? How is that precisely defined? What is the underlying statistical principle? For another instance, for the Coefficient of Variation metric, why is it defined as such? What are the theoretical benefits and properties of this measure?
3. Experiments are small-scale and the datasets are too simple, which constrains the generality of the qualitative insights.

**Questions:**

1. What is exactly new in this paper and how would that help users build better VAE?
2. For Gaussian balls, why could each code represent a stochastic neighborhood? How is that precisely defined? What is the underlying statistical principle? For the Coefficient of Variation metric, why is it defined as such? What are the theoretical benefits and properties of this measure?

---

### Official Review · Reviewer_WE5H · 2025-10-31

**Soundness:** 2
**Presentation:** 2
**Contribution:** 2
**Rating:** 4
**Confidence:** 4

**Summary:**

In the paper, the authors disentangle the components in VAE and show that KL divergence is the primary force behind forming meaningful semantic manifolds. While reparameterization helps by making latent codes continuous and Gaussian-shaped, the paper argues its more critical role lies in diversifying semantic samples, enabling effective sample-based generation. Based on these insights, the authors present a unified perspective in which both VAE and VQ-VAE are special cases governed by latent compactness enforced by KL regularization. Intriguingly, this framework also explains how VQ-VAE, despite lacking stochasticity and a continuous prior, can still generate high-quality samples. Overall, the paper provides a fresh conceptual lens on VAEs, highlighting KL divergence—not stochasticity—as the key driver of semantic structure in latent space.

**Strengths:**

This work revisits how VAEs achieve meaningful latent representations by examining the distinct roles of KL divergence and reparameterization. Rather than taking the traditional variational-inference perspective for granted, the study explores whether VAE behavior can be understood from a more empirical and geometric viewpoint. Through controlled experiments and tailored qualitative metrics, the authors provide evidence suggesting that KL divergence may be particularly important for encouraging organized semantic structure in latent space, while reparameterization appears to support stable sampling and enrich latent coverage. Although not intended as a definitive theoretical explanation, this empirical analysis offers a complementary perspective that may help broaden our understanding of latent-space behavior in VAEs and related models such as VQ-VAE.

**Weaknesses:**

1. The paper lacks theoretical justifications that supports their hypothesis. They use a small amount of visualization on small dataset to showcase and support their hypothesis.

2. They claim their method is connected to VQ-VAE and unifires VQ-VAE, whereas to me, it seems more like a combination with VA-VAE rather than unification.

3. The empirical study does not provide any ablation study to understand the strength of the variance or the regularizer, and fails to provide convincing evidence of the advantage of the proposed method.

4. Generally I feel like the contribution of the paper is below the acceptance bar of ICLR.

**Questions:**

Please see above for weakness questions.

---

### Official Review · Reviewer_cNSf · 2025-11-01

**Soundness:** 2
**Presentation:** 1
**Contribution:** 1
**Rating:** 2
**Confidence:** 4

**Summary:**

The papers presents an empirical investigation of  the effects of (1) KL regularization and (2) reparameterization in VAEs. The paper also presents a unified framework the generalizes both VAE and VQ-VAE.

**Strengths:**

The work addresses the important research topic of understanding the workings of VAEs.
The authors propose different metrics to measure the characteristics of VAEs.

**Weaknesses:**

Further proofreading is required: Mainly (1) the axes in the figures are not consistent (Figure 5, 2B), and the claims are hard to understand (see below).
Line 215: The authors write "or does it also stem from the increased density of semantic definition that more regions of the space are actively used?" It's not clear what is meant here.
Line 243: "Figure 4B ...... produces incoherent and meaningless outputs," I don't believe this is clearly shown in the figure, many examples resemble MNIST digits. can you be more precise to what the reader should look for?
Line 249: the authors write "This demonstrates that reparameterization does more than just regularize the latent space: it effectively enriches the set of defined points, acting as a contractual mechanism between the encoder and decoder." I don’t see how this follows from figure 4a. Does it not just means the new decoder is robust to small perturbations?
Section 3.1 L265: Can you explain what the "Dynamic Latent Coverage" metric? what is the formula and exactly how it is related to the prior work?
Section 3.1 seems out of place, under Section 3: VQ-VAE
Section 3.2 and Figure 6: The authors write "Model Architecture. Path A: for our experiment, where the encoder output is regularized toward the nearest code in a learnable codebook. Path B: VQ-VAE, which uses discrete latent codes." I don't understand the difference here! In VQ-VAE also the encoder output is regularized toward the nearest code in a learnable codebook.

**Questions:**

Please address the weaknesses above.

---

### Note · Authors · 2025-11-12

I have read and agree with the venue's withdrawal policy on behalf of myself and my co-authors.